# Comparing self-reported and O\*NET-based assessments of job control as predictors of self-rated health for non-Hispanic whites and racial/ethnic minorities

Kaori Fujishiro[1]*, Franziska Koessler[2,3]

1 Division of Field Studies and Engineering, National Institute for Occupational Safety and Health (NIOSH), Centers for Disease Control and Prevention (CDC), Cincinnati, Ohio, United States of America, 2 The "Good Work" Program, WZB Berlin Social Science Center, Berlin, Germany, 3 Department of Psychology, Humboldt University of Berlin, Berlin, Germany

* kfujishiro@cdc.gov

## Abstract

The Occupational Information Network (O\*NET) database has been used as a valuable source of occupational exposure information. Although good agreement between O\*NET and self-reported measures has been reported, little attention has been paid to O\*NET's utility in racially/ethnically diverse samples. Because O\*NET offers job-level information, if different racial groups have different experiences under the same job title, O\*NET measure would introduce systematic measurement error. Using the General Social Survey data ($n$ = 7,041; 437 occupations), we compared self-report and O\*NET-derived measures of job control in their associations with self-rated health (SRH) for non-Hispanic whites and racial/ethnic minorities. The correlation between self-report and O\*NET job control measures were moderate for all gender-race groups (Pearson's $r$ = .26 - .40). However, the logistic regression analysis showed that the association between O\*NET job control and SRH was markedly weaker for racial/ethnic minorities than for non-Hispanic whites. The self-reported job control was associated with SRH in similar magnitudes for both groups, which precluded the possibility that job control was relevant only for non-Hispanic whites. O\*NET may not capture job experience for racial/ethnic minorities, and thus its utility depends on the racial/ethnic composition of the sample.

## Introduction

The need has been growing for work characteristics to be incorporated into population health research, especially to address health inequity [1–4]. Work can affect health as a source of resources (e.g., income, medical insurance, social status, opportunities for self-actualization) as well as a source of harmful exposures (e.g., physical, chemical, biological hazards, job stress). Moreover, society's sorting mechanisms—such as the quality of schooling, stereotyping, and discrimination—shape the access to health-enhancing jobs and the ability to avoid health-

**Data Availability Statement:** The GSS data are publicly available from the General Social Survey Data Explorer: https://gssdataexplorer.norc.org/variables/vfilter. Interested researchers can create a

user profile and select the desired data to download. For this study, we downloaded the following variables: HEALTH1, LEARNNEW, MYSKILLS, OPDEVEL, WORKDIFF, WKDECIDE, WKFREEDM, ID_, WTSSALL, YEAR, SEX, RACE, HISPANIC, ETHNIC (as backup if HISPANIC was missing), AGE, WRKSTAT, OCC10. The O*NET database is publicly available from the O*NET Resource Center: https://www.onetcenter.org/database.html. We used items 4.C.3.b.8, 4.C.3.b.2, and 4.A.2.b.3. Because GSS provides the Census 2010 Occupation Codes but O*NET is based on the Standard Occupation Code (SOC), a crosswalk is necessary. The modified O*NET database (version 24) with the Census 2010 Occupation codes is available from NIOSH. Please contact Dr. Leslie MacDonald (lmacdonald@cdc.gov).

**Funding:** FK was a research fellow supported by the Hans-Böckler Foundation, Düsseldorf, Germany for the duration of this work. The funder had no role in study design, data collection and analysis, decision to publish, or preparation of the manuscript.

**Competing interests:** The authors have declared that no competing interests exist.

damaging jobs. The resulting pattern of who does what kind of work contributes to health inequity. Despite the complexity and importance of work in creating health and health inequity, work has seldom been addressed in a systematic way in population health research [3, 4]. In this paper, we propose that the Occupational Information Network (O*NET) database is a potentially useful tool to address work as a crucial factor that influences population health and health inequity.

## Potential for O*NET in population health research

One practical reason for work to be under-researched may be limited data availability. Collecting meaningful occupational characteristics in an analyzable form is a resource-intensive process, and as a result occupational data are often limited in large-scale cohort studies and population health surveillance programs. However, many data sources include standard codes for job titles, such as the Census Occupation Codes and Standard Occupational Classification (SOC). These codes can be used to link health data to the Occupational Information Network (O*NET) database [5]. Compiled by the US Department of Labor, the O*NET database contains a wide range of job characteristics for nearly all job titles in the US civilian workforce. The characteristics include the kind of task performed (e.g., moving heavy objects), skills and abilities required on the job (e.g. memorization, finger dexterity), and the context in which the job is performed (e.g., interaction with the public, exposure to the elements). O*NET ratings have been collected on a rolling basis from current job holders and experts who are familiar with the job [5]. The Methods section below offers more details.

Although O*NET was not designed for health research specifically, it has great potential for integrating occupational perspectives in population health research and policy interventions. First, when O*NET data are linked to other population health datasets, it removes the problem of common method biases, or artificial inflation of the association between exposure and health when both are reported by the same individual [6]. Second, O*NET data describes the job rather than the worker; therefore, if certain O*NET job characteristics are identified as health-damaging, the solution is suggested also at the job level: how to design the job to be more conducive to health. If unhealthy jobs are redesigned, all current and future workers who perform these jobs will benefit. Thus, the use of O*NET has the potential to strengthen the impact of population health research on policy intervention.

## The need for validation

For population health researchers to use O*NET with confidence, they need assurance of its usefulness. To date, two studies [7, 8] have evaluated O*NET's convergent validity (i.e., the agreement between O*NET and self-reported data measuring the same construct). Both studies reported generally good agreement for psychosocial job characteristics (i.e., job demands, job control, effort, and reward in Cifuentes et al. [7]; job control in Meyer et al. [8]) and concluded that O*NET might provide a useful metric.

The current study presents further validation from a perspective of racial health inequity. Because Cifuentes et al. [7] did not consider race and Meyer et al. [8] controlled for race, these validation studies assured the use of O*NET when race was not a central component of the study hypothesis. If O*NET-derived measures are to be used in racially diverse samples and if researchers are interested in racial/ethnic differences in the relationship between work and health, further validation is needed. O*NET offers job-level information; hence, the use of O*NET assumes that all workers under the same job title have the same exposure. Yet, because O*NET data collection was not specifically designed to ensure representation of all racial/ethnic minority groups in each job [9], it is not clear if O*NET captures occupational exposure of

all racial/ethnic groups. If there are racial differences in job exposure within the same job, the use of O*NET would introduce systematic measurement error by race. Then, O*NET may not be useful and potentially detrimental to racial health inequity research.

## Racialized experience of work

Most jobs in the United States do not represent the racial composition of the US working population. For example, in 2018 African Americans account for 12% of the civilian workforce, but the proportion of Black workers on the job ranges from <0.1% (chiropractors) to 41.1% (postal service mail sorters) [10]. Workers of color are generally overrepresented in service, manufacturing, maintenance, transportation, and material-moving (e.g., warehouse) occupations whereas they are underrepresented in professional and managerial occupations [10]. While these across-job differences are well recognized, little attention has been given to within-job differences by race, which is our central focus in examining O*NET.

We were not able to find any previous study that directly assessed racial differences in task or role assignments within the same job; however, several findings in the organizational behavior literature suggest that whites and workers of color may experience jobs differently even within the same job. In their large-scale meta-analysis, McKay and McDaniel [11] reported that Black workers were consistently evaluated as less knowledgeable and poorer performers than white workers. It was not only that supervisors evaluated Black workers unfavorably, but also subordinates perceived white leaders more effective than Black leaders [12]. When there was no organizational hierarchy, team members who did the same tasks assessed Black male workers as contributing less and thus deserving less salary than white men [13].

These findings support the concept of *über discrimination*, proposed by Reskin [14] as "a meta-level phenomenon that shapes our culture, cognitions, and institutions," (p. 17). Reskin argues that racial bias favors whites and exacerbates racial disparities in virtually all domains of life. Thus, *über* discrimination "influence[s] the cultural and social contexts in which people act. [. . .] It distorts how we see others, the attributions we make about them, and our predictions of their performance" (p. 24). Accordingly, *über* discrimination may influence the way whites and people of color experience their job. For example, if supervisors perceive that Black workers are poorer performers than whites, they may supervise the former more closely and grant less autonomy to Black workers than their white colleagues. As a result, Black and white workers may have different experiences on the same job, even under the same supervisor. One study reported that among the certified nursing assistants, Blacks reported lower job control than white colleagues in the same worksite [15]. Thus, O*NET measures, captured at the job level, may not apply to all racial groups equally.

## The current study

This study investigates whether O*NET and self-reported measures of job control derive the same conclusion for both non-Hispanic whites and racial/ethnic minorities. We focus on job control because of its well-established association with health [14, 15] and also because of its use in the previous two validation studies [8, 9]. The well-established association is important because our goal is not about showing how job control is associated with health. Rather, we examine whether the association can be detected equally across racial groups when O*NET is used to capture job control. We will first demonstrate that our sample, the General Social Survey participants, replicates the previous finding of overall agreement between O*NET and self-reported measures of job control in relation to self-rated health, a robust indicator of general health [16]. Then we will explore if the pattern of association between job control and self-

rated health differs by the source of the measure (O*NET or self-report) for non-Hispanic whites and workers of color.

We fully recognize that the above argument on racialized experience within the same job can be true for gender as well: men and women on the same job may have different experiences [16]. Moreover, increasing focus on intersectionality [17] demands simultaneous consideration for race and gender in health inequity research. While we wholeheartedly agree with the approach, in this study we are constrained by the sample size even though we use a nationally representative sample compiled over five years. We stratify all analyses by gender in order to acknowledge that the O*NET validation from a racial perspective may show different patterns by gender.

## Methods

### Data sources

**O*NET.** As mentioned above, O*NET is a database of job characteristics in the US economy. Data on job tasks, activities, and working environments were collected from workers who currently held the job, and data on abilities required for the job were rated by experts who were identified through professional organizations and educational institutions related to the job. Workers were randomly selected within business establishments that were also randomly selected. Sampled workers completed a questionnaire about their job tasks and activities, how often they performed them, and how important or intense they were. The data were then aggregated to the job level and made available to the public. More details about data collection are described by the O*NET Resource Center (https://www.onetcenter.org/dataCollection.html) and the RTI International, who conducts the data collection (https://onet.rti.org/about.cfm). The data have been continuously collected since the Department of Labor launched the program in 1998, and updates are released annually. We used O*NET version 24.0, which covers 974 jobs defined by the 2010 Standard Occupation Classification (SOC) system with O*NET-specific 2-digit extension.

**General Social Survey (GSS).** Funded by the National Science Foundation, GSS is a nationally representative cross-sectional survey conducted biannually since 1972. Through in-personal interviews, respondents report their attitudes toward various aspects of US society as well as their individual behaviors and health status [18]. We used the Quality of WorkLife module (QWL, http://gss.norc.org/Pages/quality-of-worklife.aspx), which has been included in GSS every four years starting in 2002. QWL asks about physical and psychosocial working conditions, the organization of work, and work-related well-being. Respondents in this module were at least 18 years of age and worked full- or part-time, or temporarily did not work at the time of data collection (e.g., because of parental leave) [18]. Over the five waves of QWL, 7,407 respondents provided data who work in 466 of the 539 occupations defined by the 2010 Census Occupation Codes.

**Linking the datasets.** The two datasets were linked by the 2010 Census Occupation Codes. Because O*NET was organized by SOC, a more detailed system than the Census Codes, we modified the O*NET database to be compatible with the Census Occupation Codes. In this process, we first aggregated the O*NET-specific 2-digit extension to the SOC 6-digit level (e.g., 11–1011.00 "chief executives" and 11–1011.03 "chief sustainability offers" to SOC 11–1011 "chief executives"). We then assessed the match between SOC and Census Occupation Codes. If a job title was identical between the two classification systems (e.g., chief executives—SOC 11–1011, Census 0010), we carried over the O*NET ratings for the SOC to the corresponding Census Code. When multiple SOC codes were identified under one Census code (e.g., software developers—SOC 15–1132, 15–1133; Census 1020), we calculated a

weighted mean; that is, O*NET ratings were weighted by the number of workers in each of the involved SOC codes [19]. For more details, see Appendix A of Fujishiro et al. [20].

## Measures

**O*NET job control.**   We used three O*NET items to construct a job control scale (Table 1): having freedom to set goals and to make decisions as well as applying new knowledge to the job. The first two described decision authority and the third skill discretion [21]. The two decision authority questions had a 5-point response scale; the skill discretion question asked about its importance with a 5-point scale and the level of its complexity with a 7-point scale. For ease of comparisons, we first rescaled all responses to range from 0 to 100, higher scores indicating higher job control, and averaged the importance and complexity-level responses for the "new knowledge" item; then the average of the three items was calculated as the O*NET job control score for each occupation.

**Self-reported job control.**   We used the six items available in all waves of QWL: two on decision authority (i.e., freedom to decide how to do my work, take part in decision making) and four on skill discretion (e.g., keep learning new things) (see Table 1). All items had a 4-point response scale. After reversing the response scale so that a higher score indicated a higher level of job control, we calculated the mean of the six items as the self-reported job control score.

Having four items on skill discretion, the self-reported job control measure reflected the aspect more sensitively than the O*NET job control measure, which included only one skill discretion item. We conducted a sensitivity analysis with a reduced self-report job control measure, which included only two decision authority items and one skill discretion as indicated in Table 1.

**Demographic characteristics.**   We retrieved the respondents' age in years and their gender as either "male" or "female" from GSS. Race was operationalized as being non-Hispanic white versus being any other race/ethnicity based on two items: "Are you Spanish, Hispanic, or Latino/Latina?" (yes/no) and "What is your race?" (white, Black, and other). The frequency of response is shown in S1 Table. Although we recognize that the experience of belonging to a minority racial/ethnic group is by no means homogeneous across groups, the sample size does not allow for further distinctions among them.

**Self-Rated Health (SRH).**   Respondents answered the question "Would you say in general your health is Excellent, Very good, Good, Fair, or Poor?" This single-item measure has shown strong predictive validity for mortality and morbidity [22]. The responses were dichotomized (i.e., fair/poor = 1, all other = 0) for logistic regression analysis.

## Analytic sample

After the GSS and O*NET datasets were merged, we had 7,407 respondents in 466 occupations. Of those, we removed 235 respondents because O*NET did not provide any information for their occupations (i.e., five military occupations, 24 jobs that are not specific, e.g., "office support workers, all other"). Of the remaining 7,172, those who had missing data on more than one item in the self-reported job control measure were removed ($n$ = 109). In addition, 18 who did not provide age, and one who did not provide the Hispanic ethnicity information were removed. The final dataset consisted of 7,041 workers in 437 jobs (see Table 2 for the gender and racial breakdown), representing 95.1% of all respondents of QWL compiled over five years (2002, 2006, 2010, 2014, and 2018). Women accounted for 51% of the sample, and non-Hispanic whites 70%. The average age was 42.2 years ($SD$ = 13.4). Poor/fair health was reported by 14%.

**Table 1. Available O*NET and QWL items that address job control.**

| O*NET | | Self-report (QWL) | |
|---|---|---|---|
| Item [element ID] | Response options | Item | Response options |
| *Decision authority* | | | |
| How much freedom do you have to determine the tasks, priorities, or goals of your current job? [4.C.3.b.8] | 1 = No freedom<br>2 = Very little freedom<br>3 = Limited freedom<br>4 = Some freedom<br>5 = A lot of freedom | I am given a lot of freedom to decide how to do my own work.[3] | 1 = Very true<br>2 = Somewhat true<br>3 = Not too true<br>4 = Not at all true |
| In your current job, how much freedom do you have to make decisions without supervision? [4.C.3.b.2] | 1 = No freedom<br>2 = Very little freedom<br>3 = Limited freedom<br>4 = Some freedom<br>5 = A lot of freedom | In your job, how often do you take part with others in making decisions that affect you? [3] | 1 = Often<br>2 = Sometimes<br>3 = Rarely<br>4 = Never |
| *Skill discretion* | | | |
| UPDATING AND USING RELEVANT KNOWLEDGE: (defined as) Keeping up-to-date technically and applying new knowledge to your job. [4.A.2.b.3]<br><br>How important is UPDATING AND USING RELEVANT KNOWLEDGE to the performance of your current job?<br><br>What level of UPDATING AND USING RELEVANT KNOWLEDGE is needed to perform your current job? | *How important*<br>1 = Not important[1]<br>2 = Somewhat important<br>3 = Important<br>4 = Very important<br>5 = Extremely important<br><br>*What levels is needed*:<br>Low (1) to High (7)[2] | My job requires that I keep learning new things. [3] | 1 = Strongly Agree<br>2 = Agree<br>3 = Disagree<br>4 = Strongly disagree |
| | | My job lets me use my skills and abilities | 1 = Strongly Agree<br>2 = Agree<br>3 = Disagree<br>4 = Strongly disagree |
| | | I have an opportunity to develop my own special abilities | 1 = Strongly Agree<br>2 = Agree<br>3 = Disagree<br>4 = Strongly disagree |
| | | I get to do a number of different things on my job | 1 = Very true<br>2 = Somewhat true<br>3 = Not too true<br>4 = Not at all true |

[1] When the importance was 1, O*NET set the level to be 0.

[2] The scores from the two sets of responses were used to calculate the weighted average score as $((Importance - 1)/(5-1) + Level/7)/2$. For example, if Importance = 3 and Level = 4, the weighted average score was $((3-1)/(5-1) + 4/7)/2 * 100 = 50$. The weighted average ranges from 0 to 100.

[3] Included in the reduced job-control measure.

## Statistical analysis

We examined the psychometric properties of the self-reported job control measure, using Cronbach's alpha to assess overall internal consistency and intraclass correlation (ICC1) to

**Table 2. Correlation coefficients between O\*NET and self-report job control measures.**

|  | n | Number of jobs held | Individual level (Pearson's *r*) | Cross level (*β*) |
|---|---|---|---|---|
| All respondents | 7041 | 437 | .35 | .32 |
| Men |  |  |  |  |
|   White men | *2459* | 339 | .40 | .38 |
|   Non-white men | *937* | 256 | .33 | .30 |
| Women |  |  |  |  |
|   White women | *2449* | 309 | .34 | .34 |
|   Non-white women | *1196* | 226 | .26 | .25 |

All coefficients are significant at p < .001. *β* = regression coefficient from a mixed linear model with self-reported job control as the dependent variable and O\*NET measure as the independent variable with occupation as the random effect.

quantify the proportion of variance attributable to the job. We also examined correlations between self-report and O\*NET measures at individual level and cross level (i.e., self-report at the individual level, O\*NET at the job level). Because O\*NET measures are commonly used at the individual level [23], we calculated Pearson's *r* by treating them as individual-level measures. The cross-level correlation was examined by fitting a linear mixed model with the self-report measure as the dependent variable predicted by the O\*NET measure with a random effect for occupation. The standardized regression coefficient (β) for the O\*NET measure was interpreted as the cross-level correlation coefficient.

We then compared self-reported and O\*NET-derived measures in their associations with SRH. We modeled each measure in individual-level logistic regression using the full sample, adjusted for age, race/ethnicity, gender, and GSS data collection year. Then we repeated the analysis but accounted for the hierarchical nature of data (i.e., workers nested in jobs) by applying multilevel modelling. This was to confirm that our sample would replicate the same good agreement between O\*NET and self-report, as shown in the previous validation studies [7, 8].

In exploring O\*NET and self-report agreement by race, we stratified the sample by gender so as not to obscure potential gender differences. For the O\*NET job control measure, we fitted mixed effects logistic regression with random effects for occupation to estimate the odds ratio of reporting poor/fair health associated with job control. For the self-reported measure, we used regular logistic regression as is customary in the literature. We accounted for the race of the worker (non-Hispanic white = 1, otherwise = 0; Model 1), and then included the interaction between job control and race (Model 2) to examine if the association between job control and SRH depended on the worker's race. All models were adjusted for age and survey year. Predicted probability of reporting poor/fair health, based on Model 2, was presented by gender and race so that the pattern of association between job control and SRH could be seen clearly.

In regression analyses, we standardized all independent variables with the mean of zero and standard deviation of 1 (O\*NET score at the job level, all others at the individual level) and applied sampling weights provided in GSS. All data management and processing were conducted in *R*Studio [24]. Multilevel analysis was conducted using the *lme4*-package [25].

## Results

### Descriptive findings on job control measures

**Psychometric properties of self-reported measures.** For the full sample, the self-reported job control measure had adequate internal consistency (Cronbach's alpha = 0.70). ICC1 by the

**Table 3. Odds of reporting poor/fair health associated with job control after accounting for age, race, and gender.**

| Independent variable | OR (*95%CI*) | OR (*95%CI*) | OR (*95%CI*) |
|---|---|---|---|
| O*NET job control (job level) | 0.70 (0.64–0.77) | | |
| O*NET job control (individual)[1] | | 0.72 (0.67–0.76) | |
| Self-report job control (individual) | | | 0.72 (0.67–0.76) |
| Being racial/ethnic minority | 1.55 (1.33–1.80) | 1.51 (1.31–1.74) | 1.56 (1.36–1.80) |
| Being female | 1.04 (0.89–1.22) | 1.02 (0.89–1.17) | 1.02 (0.89–1.17) |

Age and GSS survey year are adjusted for in all models. [1]Standardized at the individual level ($n$ = 7,041, $N$ = 437).

job title was 0.18; i.e., 18% of the variance in the self-reported job control measure was attributable to the job title. When we excluded jobs in which our sample had less than 10 workers, Cronbach's alpha was unchanged, and ICC1 increased only slightly (19%). While reliability and ICC1 were generally good, both values were higher for non-Hispanic whites than for people of color (e.g., Cronbach's $\alpha$ = 0.72 and ICC1 = 25% vs. 0.68 and 19%). See S2 Table for the full results. Overall, the self-reported measure of job control had an adequate internal consistency, and a substantial proportion of its variance was attributable to the job.

**Correlations between O*NET and self-reported measures of job control.** For both measures of job control, scores ranged similarly between non-Hispanic whites and racial/ethnic minorities in both gender groups (See S2 Table). In the full sample, the correlations between self-reported and O*NET measures were $r$ = .35 at the individual level (i.e., using O*NET scores at the individual level) and $\beta$ = .32 cross-levels (i.e., the self-reported measure at the individual level, O*NET at the job level). However, the magnitude of correlation was smaller for racial/ethnic minorities in both genders. See Table 2 for the correlation coefficients by gender-race groups.

## Convergent validity in the full sample

Table 3 shows the results of the regression analyses for all respondents with age, gender, and race/ethnicity adjusted for. Consistent with the previous two studies [7, 8], O*NET and self-reported measures of job control showed the same conclusion—in fact, nearly identical odds ratios (ORs)—that higher job control was associated with lower odds of reporting poor/fair health after race and gender are controlled for. This was regardless of the way O*NET measures were treated at the job level or the individual level.

## Convergent validity by race

Table 4 shows the regression results comparing the associations of SRH with O*NET and self-reported job control measures. The variance in SRH attributable to job title was 15% for men and 6% for women. When the sample was stratified by gender and the model controlled for age and race (Model 1), the O*NET and self-reported measures offered the same conclusion for both genders: the higher the level of control, the lower the odds of reporting poor/fair health. When we included the interaction between job control and race (Model 2), the main effect remained significant for both measures, but the interaction was significant for O*NET whereas it was null for self-report. For example, for both white and minority men, the main effect of O*NET job control was health protective: a one-standard-deviation higher level of O*NET job control was associated with 36% lower odds of reporting poor health. However, for racial/ethnic minority men, the same one standard-deviation higher level of O*NET job control was also associated with 32% *higher* odds of reporting poor health (OR = 1.32, 95%CI:

**Table 4. The association between self-rated health (poor/fair) and job control by gender.**

| Independent variable | Model 1 | | Model 2 | |
|---|---|---|---|---|
| | OR (95%CI) | OR (95%CI) | OR (95%CI) | OR (95%CI) |
| | Men (*n* = 3,396, *N* = 379) | | | |
| Job control | | | | |
| O*NET | 0.70 (0.60–0.81) | | 0.64 (0.54–0.75) | |
| Self-report | | 0.69 (0.63–0.76) | | 0.69 (0.61–0.77) |
| Racial/ethnic minority | 1.44 (1.15–1.81) | 1.58 (1.28–1.94) | 1.58 (1.25–2.01) | 1.58 (1.27–1.97) |
| Minority x Job control | | | | |
| O*NET | | | 1.32 (1.05–1.67) | |
| Self-report | | | | 1.01 (0.83–1.23) |
| | Women (*n* = 3,645, *N* = 337) | | | |
| Job control | | | | |
| O*NET | 0.71 (0.63–0.80) | | 0.64 (0.55–0.74) | |
| Self-report | | 0.74 (0.68–0.81) | | 0.74 (0.66–0.83) |
| Racial/ethnic minority | 1.62 (1.31–2.00) | 1.55 (1.28–1.89) | 1.78 (1.42–2.22) | 1.55 (1.27–1.90) |
| Minority x Job control | | | | |
| O*NET | | | 1.29 (1.05–1.59) | |
| Self-report | | | | 1.00 (0.83–1.20) |

Age and GSS survey year are controlled for in all models.

1.05–1.67) after accounting for 58% higher odds associated with being racial/ethnic minority (OR = 1.58, 95%CI: 1.25–2.01). In other words, for minority men, the net benefit of higher O*NET job control is negligible because 36% lower odds of the main effect nearly cancelled by 32% higher odds of the interaction effect. The pattern was similar for minority women. Self-report job control did not show this pattern for either gender: the main effect was protective (OR = 0.69, 95CI: 0.61–0.77) and the minority x job control interaction was not significant.

Another way to illustrate these findings is by plotting the predicted probability of reporting poor/fair health by gender and race (Fig 1). In all panels, the lines are lower toward the right, indicating that the higher the job control, the lower the predicted probability of reporting poor/fair health. However, the two lines show different patterns across the four panels. For non-Hispanic white men, two lines virtually overlap, showing that the association between SRH and job control is the same for O*NET and self-reported measures. Non-Hispanic white women show a similar pattern. However, for racial/ethnic minorities, the self-reported measure has a steeper slope (i.e., stronger association) than the O*NET measure. In fact, for racial/ethnic minority men, the slope for the O*NET measure is nearly flat (i.e., no association). In general, the SRH-job control association is weaker for racial/ethnic minorities if the O*NET measure is used. For non-Hispanic whites, we did not see the same attenuation in the association.

## Additional analyses

Because this paper is to explore the usefulness of O*NET in racialized work contexts, we conducted some additional analyses that might help interested researchers plan their use of O*NET.

**Skill discretion is underrepresented in the O*NET measure.** The O*NET job control measure has fewer items on skill discretion than the self-reported measure. When we remove some of the skill discretion items from the self-report measure (as indicated in Table 1), the

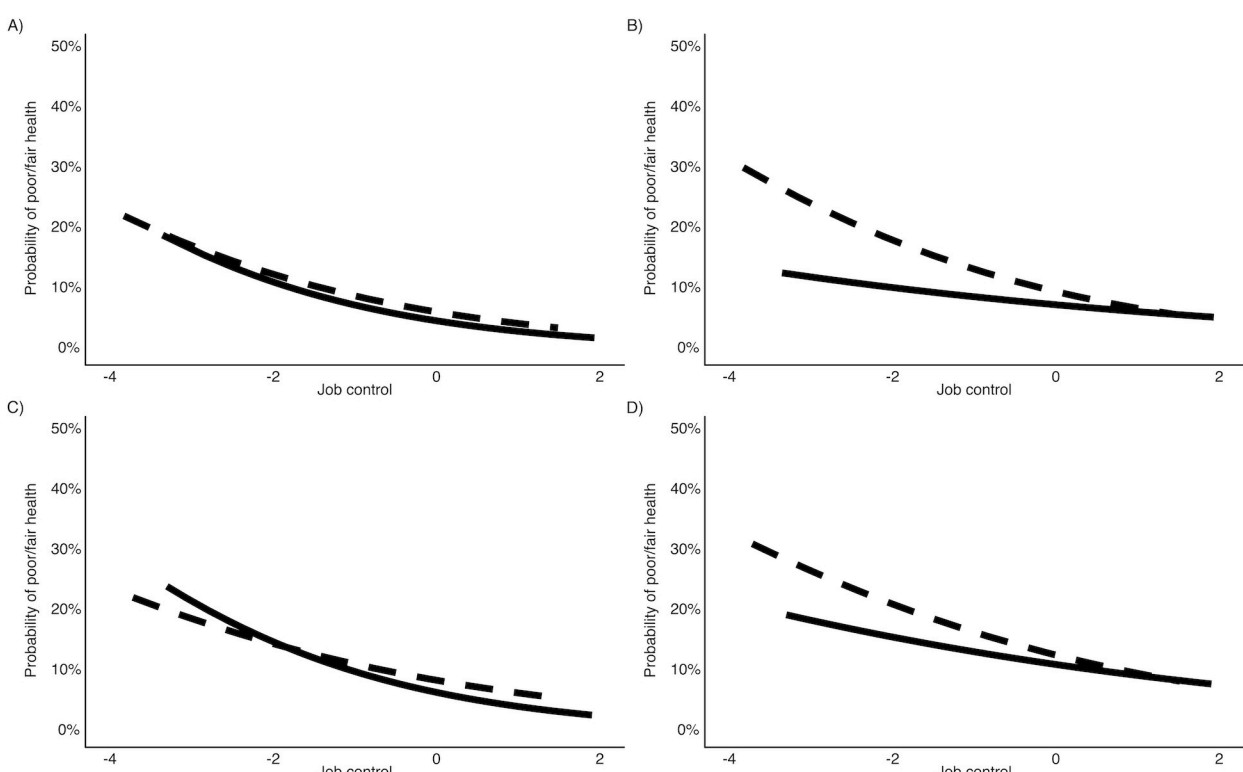

**Fig 1. Predicted probability of rating health as poor/fair by O*NET job control and self-reported job control.** Bold lines indicate O*NET job control. Dashed lines indicate self-reported job control. (A) The predicted probability of rating health as poor/fair for (A) non-Hispanic white men, (B) racial/ethnic minority men, (C) non-Hispanic white women, and (D) racial/ethnic minority women.

results are nearly identical to what we have with the original self-report measure. The full results are shown in S3 Table.

**What about job demands?** Job demands are an important dimension of work that impacts health. However, we found it difficult to operationalize with O*NET items. S4 Table shows our attempt to find corresponding items between O*NET and QWL. The time aspect of job demands was asked as mental concentration and quick response in O*NET while QWL captured it as general time pressure. The workload aspect of job demands was not included in O*NET. As expected from the apparent low face validity, the correlation between the O*NET and self-reported measures of job demands was very small and almost all non-significant (S5 Table). Therefore, we were not able to examine O*NET and self-reported job demands measures.

**Does the racial composition of the job make a difference?** In the US labor market, workers of color are overrepresented in low control jobs [26]. It is conceivable that jobs with high proportions of racial/ethnic minorities may be designed to limit the level of job control for everyone, and thus the discrepancy between O*NET and self-reported by race/ethnicity may be less. We ran the same regression model with workers in more diverse jobs (i.e., <65.7% whites, the bottom tertile of the proportion of non-Hispanic whites in the job). We were not able to stratify by gender because of the sample size constraint. The results (S6 Table) were the same as the main analysis: even in jobs with relatively high representation of racial/ethnic minorities, the main and interaction effects of O*NET job control cancelled out each other for workers of color.

## Discussion

Focusing on potential racial/ethnic differences, this study examined the convergent validity between O*NET and self-reported measures of job control in their association with SRH. The correlation between the two measures was similar across race and gender, and the full sample analysis replicated the previous validation results of good agreements between the two measures. However, our further analysis revealed that the agreement depended on the worker's race: for workers of color, O*NET job control had a markedly weaker association with SRH; for non-Hispanic whites, the attenuation was not noticeable. When job control and SRH were both self-reported, the association might be inflated because of the same source of information (i.e. common method variance) [27]; therefore, some attenuation for O*NET job control should be expected. However, different degrees of attenuation by race warrant further discussion.

The weaker association between O*NET job control and SRH for racial/ethnic minorities should not be interpreted as the lack of a well-established association for them. In fact, our data showed that self-reported job control was associated with SRH regardless of the respondent's race. The main discussion, therefore, is about the discrepancies between self-reported and O*NET-derived measures of job control.

### Subjective vs. "objective" assessments of job control

So far only few studies have compared self-report data with O*NET, but several other studies have examined the agreement between self-report and some form of objective data. For this discussion we consider data as objective if they are obtained from sources other than those whose health is studied (e.g., O*NET, expert ratings, observations by research staff). For example, the Whitehall II Study compared self-reported and objective job control assessed by personnel managers. The two sources of job control measurements were moderately correlated with each other and were associated with health outcomes in similar magnitudes [28, 29]. These are, however, results from office workers only.

In more diverse worker samples, convergent validity seems to vary. Hasselhorn and colleagues [30] examined workers recruited from the general working population of Stockholm. The correlation between self-reported and objective measures of job control was high for all jobs. However, blue-collar and skilled service workers rated their job control higher than experts whereas white-collar workers' self-reported scores were similar to expert ratings. Waldenström and Härenstam [31] reported that the discrepancy between self-report and objective measures depended on gender and the level of job control itself: among workers who reported high levels of job control, experts observed high level of job control only in men and not in women. These studies suggest that when a study sample includes a wide range of jobs and both genders, findings based on objective measures would more closely align with self-report of male and white-collar workers compared with others.

Taken together, the agreement between self-reported and objective assessments of job control is a highly complex issue that depends on gender, type of the job, and even the level of job control itself. Our findings add to this knowledge by highlighting the influence of workers' race: if O*NET job control is used in a racially diverse sample, the more diverse the sample is, the weaker the overall association may be between job control and general health. Similar attenuation may exist for other O*NET-based exposure measures and different health outcomes. By using O*NET as a substitute for self-reported data without considering the diversity in the sample, researchers might prematurely conclude that work characteristics are not associated with health. The more diverse the worker sample is, the more likely this premature

conclusion might occur. Then, work characteristics that are common in jobs with high proportions of racial/ethnic minorities may be overlooked as health determinants.

## Limitations and future directions

Because our analysis is narrowly focused on job control and self-rated health, we first suggest conducting more validation studies of O*NET from the perspective of health inequalities. Different types of exposure, such as noise and ergonomic demands, may be validated by comparing O*NET, self-report, and direct measurements (e.g., decibel meter, lumbar motion monitor). Another limitation is our use of occupational coding systems. Although O*NET distinguishes nearly 1,000 jobs, because our source of the self-rated data, GSS, uses the Census 2010 occupation codes, we had to aggregate the O*NET job titles to the Census code, which had 539 job titles. This process might have obscured fine differences among jobs distinguished by O*NET but combined in Census codes. Using self-reported data with SOC may better clarify how O*NET and self-report data agree with each other. Because GSS did not oversample racial/ethnic minorities, the sample size prohibited us from focusing on specific racial/ethnic groups other than non-Hispanic whites. Different groups may experience different dynamics in the workplace, which might influence the agreement between self-report and O*NET measures. Investigating this aspect is particularly important in addressing health inequalities from an occupational perspective.

While we wait for more O*NET validation studies, we suggest two strategies to researchers who wish to explore job characteristics and health using O*NET. First, O*NET measures should be modeled at the job level. The common practice of assigning O*NET data to the individual implicitly endorses the assumption that O*NET would replace self-reported data. Our findings, along with some previous studies [28–31], suggest that subjective and objective measures of job characteristics may capture somewhat different constructs depending on individual characteristics. By modeling O*NET measures at the job level, we can avoid blurring individual- and job-level effects. If data at hand do not allow multi-level modeling, implications of applying O*NET data to individuals should be discussed.

Second, coefficients for gender and racial/ethnic groups should be estimated separately. This can be accomplished by including cross-level interactions between O*NET variables and the race/ethnicity of the worker. Alternatively, one can conduct race-gender stratified analyses and compare the pattern of associations across groups. In some cases, such as structural equation modeling [32] and causal mediation analysis [33], measurement error could be specified and manipulated in the model. This could be used to evaluate the robustness of the finding (i.e., what is the magnitude of measurement error that would erase the observed association between O*NET variables and health? Would the magnitude differ by race and gender?) These strategies will acknowledge the possibility that O*NET variables may have different associations with health by race/ethnicity and gender. Thoughtful discussion based on these findings may generate further research questions regarding gendered and racialized experiences in the workplace and their health implications.

## Conclusions

O*NET has a potential to bridge occupational and population health research. Such an effort will illuminate work as a health determinant beyond work-related injury and illness. Recognizing this potential, many countries have been developing similar databases of job characteristics such as Germany [34], France [35], the Netherlands [36], Finland [37], and Australia [38]. All these countries have increasingly diverse working populations, and health inequality is an important social issue. Therefore, an urgent need exists for developing databases that reflect

the diverse workforce. To take full advantage of rich job-level data, we must understand the specific implications for using O*NET or similar databases, especially when our interests are investigating health inequalities. Used with caution, O*NET may offer great opportunities to advance our understanding of health inequalities from an occupational perspective, which may lead to policy-level solutions for those inequalities.

## Supporting information

**S1 Table. Frequencies of responses to Hispanic ethnicity and race questions by gender (n = 7407).**
(DOCX)

**S2 Table. Psychometric properties and descriptive statistics of O*NET and self-reported measures of job control by gender and race.**
(DOCX)

**S3 Table. The association between self-rated health (poor/fair) and job control by gender.**
(DOCX)

**S4 Table. Available O*NET and QWL items that address job demands.**
(DOCX)

**S5 Table. Correlation coefficients between O*NET and self-report job demands measures.**
(DOCX)

**S6 Table. The association between self-rated health (poor/fair) and job control by racial/ethnic job composition: Workers in less white-dominated jobs (n = 2,266, N = 100).**
(DOCX)

**S1 File. Data access information.**
(DOCX)

## Acknowledgments

**Disclaimer:** The findings and conclusions in this report are those of the authors and do not necessarily reflect the official position of the National Institute for Occupational Safety and Health, Centers for Disease Control and Prevention.

## Author Contributions

**Conceptualization:** Kaori Fujishiro.

**Data curation:** Kaori Fujishiro, Franziska Koessler.

**Formal analysis:** Franziska Koessler.

**Methodology:** Kaori Fujishiro, Franziska Koessler.

**Project administration:** Kaori Fujishiro.

**Supervision:** Kaori Fujishiro.

**Visualization:** Franziska Koessler.

**Writing – original draft:** Kaori Fujishiro, Franziska Koessler.

**Writing – review & editing:** Kaori Fujishiro, Franziska Koessler.

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
