## [Decision Letter · Decision Letter 0]

6 Jul 2020

PONE-D-20-08355

Comparing self-reported and O*NET-based assessments of job control as predictors of self-rated health for non-Hispanic whites and racial/ethnic minorities

PLOS ONE

Dear Dr. Fujishiro,

Thank you for submitting your manuscript to PLOS ONE. We apologize for the delay in getting back to you; the editor was changed after its initial assignment. After careful consideration, we feel that your manuscript has merit but does not fully meet PLOS ONE’s publication criteria as it currently stands. Therefore, we invite you to submit a revised version of the manuscript that addresses the points raised during the review process, detailed below.

We look forward to receiving your revised manuscript.

Kind regards,

Maria Glymour, ScD, M.S.

Academic Editor

PLOS ONE

Journal Requirements:

Comments :

This paper will be of great use to the field. I have a few requests for clarification:

- The phrasing of the O*net skill discretion questions is unclear from the text or the table. Is a stem missing?

- Please clarify the number of individuals in each of the groups (non-hispanic white men, non-hispanic white women, minority men, minority women).

- The “minority” category here is quite large. Is it possible to evaluate with any more specificity, e.g., differentiate between Latinx and Black respondents? If not, at a minimum report the composition.

-Please emphasize the magnitude of associations and confidence intervals rather than statistical significance. For example, in lines 248-250, it would be helpful to report the main effect, CI, the interaction, CI, and then the estimated joint effect (e.g., main effect * interaction) for minorities.

- For figure 1, consider estimating cubic splines or some other extremely flexible functional form for the association between job control and self-rated health. I think this would also at least partially address the Reviewer’s insightful comments.

Reviewers' comments:

Reviewer's Responses to Questions

**Comments to the Author**

1. Is the manuscript technically sound, and do the data support the conclusions?

Reviewer #1: Yes

2. Has the statistical analysis been performed appropriately and rigorously? 

Reviewer #1: Yes

3. Have the authors made all data underlying the findings in their manuscript fully available?

Reviewer #1: Yes

4. Is the manuscript presented in an intelligible fashion and written in standard English?

Reviewer #1: Yes

5. Review Comments to the Author

Reviewer #1: Dear authors,

I believe your manuscript has the potential to contribute to our field after a major revision. Please find below several issues that you need to address to improve your paper:

1) Explain better the Significance of O*NET: In your introduction, you describe the purposes and the process of O*NET, but I encourage you to explain better the significance of this data source for research, policy, or practice. To that end, please cite how many studies have used this source recently or what are other implications of using the information available in O*NET. This can also help your case about the importance of your findings.

2) Please explain why you are only analyzing job control: I understand the significance of job control as a measure of occupational stress. However, I wondered why you did not assess job demands or job strain (the combination of high demands and low control), which are also known predictors of many adverse health outcomes. Please explain your rationale, whether you focused on job control on theoretical grounds, data availability, or other reasons

3) Provide more depth on a theoretical framework for occupational health inequities. Please provide a conceptual/theoretical framework for health inequities, including its drivers.

4) Measures of race/ethnicity: Along those lines, please give more details about the questions related to race/ethnicity in the GSS. Please also show the distribution of self-reported race/ethnicity and the rationale for creating a binary analytical variable.

5) Review of literature on job strain racial disparities. Please extend your search for studies that have examined how racial/ethnic minorities experience workplace factors such as job control et al. differentially than non-Hispanic whites.

6) Intersectionality: I agree with your decision of classification and analysis gender-stratified. Still, your discussion needs a mention of intersectionality as O*NET seems to perform worse for working women of color.

7) Finally, your conclusions suggest that workers of color rate job control differently than the professional raters from O*NET. We also know that people of color tend to be more over-represented in high-risk occupations. Can you make additional multilevel models to test whether the discrepancy between self-rated and professional-rated job control varies by occupations with a higher percentage of people of color?

Thanks, and I hope my comments have been useful.

6. PLOS authors have the option to publish the peer review history of their article (what does this mean?). If published, this will include your full peer review and any attached files.

Reviewer #1: **Yes: **David A. Hurtado

---

## [Author Response · Author response to Decision Letter 0]

14 Jul 2020

Response to Reviewers

We appreciate the Academic Editor and the Reviewer’s helpful and encouraging comments on our previous draft. In the revised manuscript, we clarified our goal, presented the conceptual rationale for the project, and emphasized the potential contribution of O*NET in population health research. We also included some additional analyses to address the concerns raised. Our point-by-point responses are below. We believe the paper has become stronger and more useful to the PLOS ONE readership. 

Academic Editor’s Comments:

This paper will be of great use to the field. I have a few requests for clarification:

1- The phrasing of the O*net skill discretion questions is unclear from the text or the table. Is a stem missing?

Response: The questions were asked part of a series in which concepts were defined, and the respondents were asked to indicate how important each concept was and what level of the concept is required on the job. We revised Table 1 to show the full text relevant to the skill discretion questions. 

2- Please clarify the number of individuals in each of the groups (non-hispanic white men, non-hispanic white women, minority men, minority women).

Response: The information is now added to Table 2. 

3- The “minority” category here is quite large. Is it possible to evaluate with any more specificity, e.g., differentiate between Latinx and Black respondents? If not, at a minimum report the composition.

Response: As shown in the new S1 Table (created in response to Reviewer 1’s comment 4), the Hispanic ethnicity and Black or “other” race are not mutually exclusive categories. Also, the sample size becomes rather small after gender stratification. We assume that a good portion of “others” are Asians, and their experience in the workplace can be as diverse (e.g., Vietnamese seafood processing workers and Indian computer engineers) as that of Latinx who identify also as white vs. Black. We recognize the crude categorization of race and ethnicity in this sample and its limitation, but unfortunately the GSS data do not allow further investigation (mentioned in lines 218-221). 

4-Please emphasize the magnitude of associations and confidence intervals rather than statistical significance. For example, in lines 248-250, it would be helpful to report the main effect, CI, the interaction, CI, and then the estimated joint effect (e.g., main effect * interaction) for minorities.

Response: We added a narrative interpretation of the regression results, including the main and interaction effects and their CIs (lines 316-327). 

5- For figure 1, consider estimating cubic splines or some other extremely flexible functional form for the association between job control and self-rated health. I think this would also at least partially address the Reviewer’s insightful comments.

Response: We tried to fit cubic splines, but it created some singularity problems. We are also not sure how splines could be applied to interaction models. In terms of racial groups having different levels of job control (Reviewer 1’s comment #7), we included a subsample analysis of those who were in less white-dominated jobs (lines 368-377, Table A6). The findings were virtually the same as the main analysis. 

Reviewer #1: Dear authors,

I believe your manuscript has the potential to contribute to our field after a major revision. Please find below several issues that you need to address to improve your paper:

1) Explain better the Significance of O*NET: In your introduction, you describe the purposes and the process of O*NET, but I encourage you to explain better the significance of this data source for research, policy, or practice. To that end, please cite how many studies have used this source recently or what are other implications of using the information available in O*NET. This can also help your case about the importance of your findings.

Response: We revised the introduction extensively. It now has a section, “potential for O*NET in population research” (lines 49 - ), which spells out potential merits for using O*NET (lines 63- 72). We did not feel that reporting the number of recent studies that have used O*NET would strengthen our point. It is still small—the Cifuentes et al. review has been cited 41 times but not all are empirical studies or about health. This underutilization is part of the motivation for our study. 

2) Please explain why you are only analyzing job control: I understand the significance of job control as a measure of occupational stress. However, I wondered why you did not assess job demands or job strain (the combination of high demands and low control), which are also known predictors of many adverse health outcomes. Please explain your rationale, whether you focused on job control on theoretical grounds, data availability, or other reasons

Response: We clarified the rationale for using job control in the introduction (lines 128-131). We also added our exploration on the job demands measure in the results section (lines 359 - 367, S4 and S5 Tables). In short, job demands measures in O*NET and QWL did not match. As self-report (as in QWL), job demand measures almost always have less-than-ideal internal consistency (Cronbach’s alpha < 0.70.). A possible reason may be that the questions are interpreted as asking about the worker’s own abilities (“Are you able to finish everything in time?”) instead of asking about work arrangements. Also possible is that workload varies more widely across workplaces through management practice and local economy, rather than across job titles. Even if O*NET offered similar items, the low internal consistency would have brought in unwanted ambiguity in our investigation of convergent validity. 

3) Provide more depth on a theoretical framework for occupational health inequities. Please provide a conceptual/theoretical framework for health inequities, including its drivers.

Response: Occupational health inequity/disparities is still not a well-defined concept (Ahonen et al. 2018), but several scholars have pointed out reasons for investigating work as an important factor in shaping health inequity. We expanded the first paragraph of the introduction to briefly summarize the view (lines 38-48). We also developed a new section on racialized work experience in the introduction (lines 93-123) to give a conceptual foundation for this study. 

4) Measures of race/ethnicity: Along those lines, please give more details about the questions related to race/ethnicity in the GSS. Please also show the distribution of self-reported race/ethnicity and the rationale for creating a binary analytical variable.

Response: The distribution of the response to the two questions is now shown in S1 Table. GSS does ask more detailed questions about racial identity and ethnic origin, including multi-racial/ethnic identities. However, the more detailed data we use, the more complex it becomes (rightly so) to characterize people with categories of race and ethnicity. Also, the sample sizes for each group becomes small especially after stratified by gender. In this study, our focus is to examine if the use of O*NET introduces systematic measurement error to workers of color in comparison with whites. While we acknowledge the heterogeneity of racialized work experiences, the current data does not allow us to investigate any further (stated in lines 218-221). 

5) Review of literature on job strain racial disparities. Please extend your search for studies that have examined how racial/ethnic minorities experience workplace factors such as job control et al. differentially than non-Hispanic whites.

Response: When we started this project, we were surprised to discover that within-job racial differences have not been directly studied. In the new section of racialized work experience (lines 93-123), we present our argument for whites and workers of color to have different experience on the same job. 

6) Intersectionality: I agree with your decision of classification and analysis gender-stratified. Still, your discussion needs a mention of intersectionality as O*NET seems to perform worse for working women of color.

Response: By stratifying the sample by gender, we allowed the racial differences of O*NET performance to vary by gender. The results, however, showed similar racial differences in both genders. At least on the construct of job control, being a person of color seems to be more important and being a woman. We recognize the importance of intersectionality and expressed our view in the introduction (lines 137-144). 

7) Finally, your conclusions suggest that workers of color rate job control differently than the professional raters from O*NET. We also know that people of color tend to be more over-represented in high-risk occupations. Can you make additional multilevel models to test whether the discrepancy between self-rated and professional-rated job control varies by occupations with a higher percentage of people of color?

Response: The three O*NET ratings we used for this analysis are provided by current workers on the job. Presumably, the racial/ethnic composition is reflected in the sample, but there was no specific effort made to ensure the proportional representation (now stated in lines 87-89). We added an additional analysis of limiting the sample to workers who have jobs with more diverse racial/ethnic composition (lines 368-377, S6 Table). These jobs still have mostly white majority. The results are the same as the main analysis: even in jobs with higher proportions of people of color, O*NET performed better for whites.

---

## [Editor Report · Decision Letter 1]

20 Jul 2020

Comparing self-reported and O*NET-based assessments of job control as predictors of self-rated health for non-Hispanic whites and racial/ethnic minorities

PONE-D-20-08355R1

Dear Dr. Fujishiro,

We’re pleased to inform you that your manuscript has been judged scientifically suitable for publication and will be formally accepted for publication once it meets all outstanding technical requirements.

Kind regards,

Maria Glymour, ScD, M.S.

Academic Editor

PLOS ONE

Additional Editor Comments (optional):

Thanks for the nice and thoughtful revisions.
---

## [Editor Report · Acceptance letter]

24 Jul 2020

PONE-D-20-08355R1 

Comparing self-reported and O*NET-based assessments of job control as predictors of self-rated health for non-Hispanic whites and racial/ethnic minorities 

Dear Dr. Fujishiro:

I'm pleased to inform you that your manuscript has been deemed suitable for publication in PLOS ONE. Congratulations! Your manuscript is now with our production department. 

Kind regards, 

on behalf of

Dr. Maria Glymour 

Academic Editor

PLOS ONE